# Prevalence and Characteristics of Isolated Nocturnal Hypertension and Masked Nocturnal Hypertension in a Tertiary Hospital in the City of Buenos Aires

**DOI:** 10.3390/diagnostics13081419

**Published:** 2023-04-14

**Authors:** Jessica Barochiner, Romina R. Díaz, Rocío Martínez

**Affiliations:** 1Hypertension Section, Internal Medicine Department, Hospital Italiano de Buenos Aires, Buenos Aires C1199, Argentina; rominar.diaz@hospitalitaliano.org.ar (R.R.D.); rocio.martinez@hospitalitaliano.org.ar (R.M.); 2Instituto de Medicina Traslacional e Ingeniería Biomédica (IMTIB), UE de Triple Dependencia CONICET-Instituto Universitario del Hospital Italiano (IUHI)-Hospital Italiano (HIBA), Buenos Aires C1199, Argentina

**Keywords:** isolated nocturnal hypertension, masked hypertension, ambulatory blood pressure monitoring, predictors

## Abstract

Isolated nocturnal hypertension (INH) and masked nocturnal hypertension (MNH) increase cardiovascular risk. Their prevalence and characteristics are not clearly established and seem to differ among populations. We aimed to determine the prevalence and associated characteristics of INH and MNH in a tertiary hospital in the city of Buenos Aires. We included 958 hypertensive patients ≥ 18 years who underwent an ambulatory blood pressure monitoring (ABPM) between October and November 2022, as prescribed by their treating physician to diagnose or to assess hypertension control. INH was defined as nighttime BP ≥ 120 mmHg systolic or ≥70 diastolic in the presence of normal daytime BP (<135/85 mmHg regardless of office BP; MNH was defined as the presence of INH with office BP < 140/90 mmHg). Variables associated with INH and MNH were analyzed. The prevalences of INH and MNH were 15.7% (95% CI 13.5–18.2%) and 9.7 (95% CI 7.9–11.8%), respectively. Age, male sex, and ambulatory heart rate were positively associated with INH, whereas office BP, total cholesterol, and smoking habits showed a negative association. In turn, diabetes and nighttime heart rate were positively associated with MNH. In conclusion, INH and MNH are frequent entities, and determination of clinical characteristics such as those detected in this study is critical since this might contribute to a more appropriate use of resources.

## 1. Introduction

Hypertension is currently considered the leading cause of death and disability in that as a risk factor, it underlies cardiovascular disease [1]. For many years, the cornerstone on which the diagnosis and treatment of hypertension have been based was the measurement of blood pressure (BP) in the office. However, it has several drawbacks such as the alarm reaction, observer bias, and poor reproducibility, which has contributed, in the recent history of medicine, to the rise of standardized BP measurement techniques outside the office, such as ambulatory blood pressure monitoring (ABPM) and more recently home blood pressure monitoring (HBPM) [2,3]. ABPM consists of placing a monitor that automatically records BP at regular intervals for 24 h, including sleep.

On the other hand, isolated nocturnal hypertension (INH) is defined as an abnormally elevated nocturnal BP in the presence of normal daytime ambulatory BP [4,5,6]. This entity is associated with increased cardiovascular risk in terms of target organ disease and an increased risk of cardiovascular events and mortality [7,8,9,10].

Of note, masked nocturnal hypertension (MNH), i.e., controlled office BP with INH, is a subtype of masked hypertension [11], more difficult to detect, and insufficiently studied both in treated and untreated subjects.

Office BP does not allow the detection of any of these conditions, as they can only be diagnosed by ABPM or HBPM and especially the former since home BP monitors that can be programmed to perform measurements during sleep exist but are not very widely used [12,13,14].

Given that ABPM is expensive and not quite accessible [15,16,17], it is crucial to establish that both INH and MNH are frequently sufficient for attempting to detect them, and it is also very important to determine characteristics of these conditions that would make performing an ABPM worth it. Moreover, INH and MNH have been studied primarily in Asian populations [11,18]. Data on other populations are scarcer. As a consequence, in this study, we aimed to determine the prevalence of INH and MNH and its associated characteristics in a tertiary hospital in the city of Buenos Aires.

## 2. Materials and Methods

This was a cross-sectional study that consecutively included patients aged 18 years or older who performed an ABPM between 1 October and 30 November 2022 in the Hypertension Section of Hospital Italiano de Buenos Aires. The patients underwent an ABPM as prescribed by their treating physician to diagnose hypertension or to assess its control. Therefore, subjects could have any BP level and be or not be under antihypertensive treatment. The study protocol was approved by the local ethics committee (Comité de Ética de Protocolos de Investigación (CEPI), approval #6464). The patients duly authorized the use of the information in their medical records under the protection of their confidentiality through informed consent. Duplicate ABPMs as well as ABPMs with less than 70% of attempted measurements were excluded from the analysis.

Ambulatory BP was measured for 24 h at 15 min intervals during the day (6 a.m.–10 p.m.) and 20 min intervals during the night (10 p.m.–6 a.m.) using validated devices, i.e., Spacelabs 90207 and Spacelabs 90217 (SpaceLabs Inc., Redmond, WA, USA) placed on the non-dominant arm. Patients were provided with a log where they had to register sleep and wake-up time, activities performed during the day, and the presence of symptoms. Sleep duration was then calculated, and daytime and nighttime were defined according to the diary. 

Office BP was measured with the same device prior to performing the ABPM. Measurements were made with the patient in a sitting position, with his/her back supported, and without crossing the legs; with the arm uncovered, supported, and at heart level; and without speaking.

INH was defined as nighttime BP of ≥120 mmHg systolic and/or ≥70 mmHg diastolic in the presence of a normal daytime BP (<135 mmHg systolic and <85 mmHg diastolic) regardless of office BP [11]. MNH was defined as the presence of INH with office BP < 140/90 mmHg. In turn, isolated diurnal hypertension was defined as daytime BP ≥ 135 and/or 85 mmHg and nighttime BP < 120/70 mmHg, day-night hypertension as daytime BP ≥ 135 and/or 85 mmHg and nighttime BP ≥ 120 and/or 70 mmHg, and normal day and nighttime BP as daytime BP < 135/85 and nighttime BP < 120/70 mmHg < 120/70 mmHg.

The medical records of all patients were reviewed to gather data regarding risk factors (diabetes and smoking habits), a history of cardiovascular disease (coronary heart disease and cerebrovascular disease), and the use of antihypertensive drugs. Laboratory data (fasting plasma glucose (FPG), total cholesterol, and serum creatinine) from 12 months before ABPM were also gathered from the medical records. Weight and height were also extracted, and body mass index (BMI) was calculated as weight/height^2^ (kg/m^2^).

Sample size was estimated assuming a prevalence of INH of 12.9% and according to previous data from a similar population [19]. For a precision of ±3% and a confidence level of 95%, the minimum number of patients to be recruited was calculated as 480. We assumed a possible loss rate of 10%, and therefore, the final number of participants to be recruited was at least 528. 

Results are reported as the percentage, mean ± standard deviation, or median and interquartile range, according to the data distribution. The characteristics of patients with INH, isolated diurnal hypertension, day-night hypertension, and normal day and nighttime BP were compared using one-way ANOVA test for continuous variables and the chi-square test for categorical variables. In turn, subjects with and without MNH were compared using the *t*-test for continuous variables and the chi-square test for categorical variables. A two-sided *p*-value of <0.05 was considered statistically significant. Statistical analyses were performed using STATA 14 (StataCorp, LLC, College Station, TX, USA).

## 3. Results

We included 958 patients in the study. After excluding duplicate ABPM and reports with less than 70% of attempted measurements, 917 participants remained for analysis (Figure 1).

Mean age was 65.1 (±14.7) years, and 43.3% were men. A total of 73.9% of participants were taking antihypertensive drugs, with an average of two drugs per patient. Patient characteristics are depicted in Table 1, whereas office and ambulatory BP profiles are depicted in Table 2, and the type of antihypertensive treatment in those medicated is depicted in Figure 2. 

The prevalence of masked hypertension, i.e., adequate office BP control (<140/90 mmHg) and elevated BP on ABPM (diurnal BP ≥ 135 and/or 85 mmHg or nighttime BP ≥ 120 and/or 70 mmHg), was 16.2% (95% CI 14–18.8%). The prevalences of INH and MNH were 15.7% (95% CI 13.5–18.2%) and 9.7 (95% CI 7.9–11.8%), respectively. When performing a sensitivity analysis and excluding subjects with a history of cardio or cerebrovascular disease, the results did not significantly change: the prevalence of INH was 15% (95% CI 12.7–17.6%), and the prevalence of MNH was 9.6% (95% CI 7.7–11.8%). There were 149 subjects with masked hypertension, i.e., patients with controlled office BP but elevated BP in the ABPM. Among this group, 89 patients had masked hypertension only at the expense of INH (59.7%).

In turn, the prevalences of isolated diurnal hypertension and day-night hypertension are depicted in Figure 3, whereas the level of nighttime BP in the three categories (INH, isolated diurnal hypertension, and day-night hypertension) is compared in Figure 4 and Figure 5 for systolic and diastolic BP, respectively.

In bivariate analyses, patients with INH were significantly older, with a lower prevalence of smoking habits, a predominance of men, a higher prevalence of diabetes, a higher ambulatory heart rate, and a tendency to be more medicated for hypertension as compared with subjects with normal day and nighttime BP. They also had a lower cholesterol level and a lower office systolic and diastolic BP as compared with subjects with isolated diurnal hypertension and day and nighttime hypertension (Table 3).

In turn, patients with MNH tended to be more medicated for hypertension, had a higher prevalence of diabetes, and had a higher nighttime heart rate than subjects without MNH (Table 4).

### Analysis of Untreated and Treated Subjects as Separate Groups

When analyzing treated (*n* = 678) and untreated subjects (*n* = 239) separately, the prevalences of INH and MNH were 17.1% (95% CI 14.5–20.1%) and 10.8% (95% CI 8.6–13.3%), respectively, in the former and 11.7% (95% CI 8.2–16.5%) and 6.7% (95% CI 4.1–10.7%), respectively, in the latter. Table 5, Table 6, Table 7 and Table 8 show the comparison between subjects with INH, isolated diurnal hypertension, day and nighttime hypertension, and day and nighttime normal BP and a comparison between subjects with and without MNH in untreated and treated participants, respectively.

Given that the study was designed and that sample size calculation was performed for the total population as a whole, when analyzing the two subgroups separately, some of the variables lost statistical significance, probably as a result of a loss in statistical power. For instance, in untreated subjects, only male sex and office BP were associated with INH, and no variable showed association with MNH, whereas in treated subjects, smoking habits, total cholesterol, office BP, and nighttime heart rate remained associated with INH and with the addition of beta blocker use (use of antihypertensive medication was analyzed in this subgroup); and diabetes and nighttime heart rate remained associated with MNH. 

## 4. Discussion

In our study, we found that both INH and MNH are frequent entities in Argentinean patients and that their associated characteristics are age, male sex, smoking habits, total cholesterol, ambulatory heart rate and office BP and nighttime heart rate, respectively, with a statistical tendency for being medicated for hypertension.

Our prevalence of INH is consistent with other studies: Salazar et al. found a prevalence of 12.9% in a similar Argentinean population that was slightly younger than ours [19]. In a multiethnic international database on ABPM, the prevalence of INH was a little lower, between 6 and 10.9% [20], whereas in a Chinese study [18] and in a Korean study [11], the prevalences of INH were 11.3 and 22.8%, respectively. Interestingly, in our study, INH was more prevalent than isolated diurnal hypertension. This finding is in line with a previous study conducted by Omboni et al. in 20,773 Italian persons who underwent ABPM in local pharmacies, where they found that INH was more common (16%) than isolated daytime hypertension (9%) [21]. The differences in prevalence among studies may be explained by ethnic diversity, different patient recruitment methods, and different definitions of nighttime. For example, some studies used fixed time intervals, whereas others use the patient’s diary to establish sleep duration [11,19]. Remarkably, in our study, 59.7% of masked hypertension was explained by INH. This is not a minor issue since, among the two standardized techniques to detect masked hypertension, ABPM is the one able to assess nighttime BP. In a study conducted by Kario et al., the authors showed that systolic BP obtained by HBPM was a good predictor of cardiovascular events independent of in-office and morning in-home SBP measurement [12]. Although these and other studies demonstrate the utility of nighttime BP assessment through HBPM, it should be emphasized that these HBPM devices that can be programmed to automatically measure nighttime BP are scarce and not widely used worldwide. It seems they are currently mostly reserved for research purposes. It must be highlighted that ABPM is more expensive and less tolerated by patients than HBPM [5]. Thus, a high degree of suspicion of the presence of nocturnal hypertension and the appropriate determination of its predictors are crucial for the correct use of this little accessible resource.

Regarding MNH, information is even scarcer. We found a study conducted by Rhee et al. in which the authors evaluated the prevalence and characteristics of patients with INH in the general population in Korea [11]. Using the published results, we were able to calculate the prevalence of this entity, which was 12% (99/823), slightly higher than ours.

Several mechanisms may be responsible for nocturnal hypertension: increased sympathetic nervous system activity, hyperactivity of renin-angiotensin-aldosterone system, sodium retention, renal function impairment, etc. [22,23,24]. This proposed pathophysiology is in line with previously reported predictors of INH such as diabetes, nighttime heart rate, sleep duration, older age, serum creatinine, and masked hypertension, among others [25,26,27,28]. Our findings are also consistent with these previous results in different populations and highlight the universal application of these concepts. In particular, our finding of a negative association between office BP and INH is a reflection of the previously reported association between masked hypertension and INH [11], which is a strong call for attention to measure out-of-office BP. On the other hand, our finding of a positive association between nighttime heart rate and both INH and MNH is an indicator of activated sympathetic activity and consistent with the described proposed pathophysiology [22]. Other studies have found that short sleep duration is a predictor of INH. In fact, sleep restriction (4 h of sleep per night) for 9 days has been shown to cause persistent and significant elevation in 24 h and sleep-time BP in women [29]. In our study, we were not able to find such association. The aforementioned mechanisms constitute the background to explain the association between INH and an increased cardiovascular risk. For example, in the Pressioni Arteriose Monitorate E Loro Associazioni (PAMELA) study, INH was associated with increased carotid intima–media thickness and relative wall thickness [30]; in the Jackson Heart Study, INH was related to increased left ventricular mass compared with normotension [31], whereas in a Chinese population, Li et al. showed that INH was associated to increased arterial stiffness [32]. In turn, an analysis of the International Database on Ambulatory Blood Pressure in relation to Cardiovascular Outcomes showed that, during a median follow-up of 10.7 years of 11 population cohorts from Asia, Europe, and South America that included 8711 subjects, after adjusting for cohort, sex, age, body mass index, current smoking, alcohol intake, serum total cholesterol, history of cardiovascular disease, and diabetes mellitus, INH was significantly associated with a higher risk of all-cause mortality (hazard ratio (HR) 1.29, 95% CI, 1.01–1.65, *p* = 0.045) and all cardiovascular events (HR 1.38, 95% CI, 1.02–1.87, *p* = 0.037) [25]. Of note, further adjustment for daytime BP did not change the results. In another study conducted by Salazar et al., the authors found that, in women with high-risk pregnancies, INH predicts the subsequent development of preeclampsia [33]. These results somehow change the paradigm of hypertension management since they highlight the importance of actively screening for high nighttime BP.

The case of MNH is of particular interest since neither office BP nor usual HBPM are able to detect it. ABPM is not only a limited resource but is also underutilized by primary care physicians. In the MAMPA study, for example, the degree of knowledge and management of automated devices for office BP measurement, HBPM, and ABPM was examined in primary care in Spain [34]. For that purpose, an online self-administered survey was sent to 2221 primary-care physicians working across Spain. Although almost half (47.5%) the participants considered ABPM to be the better method to diagnose hypertension, 51% only recommended it occasionally. The reasons given by physicians for this disparity were the lack of access to this device (69.8%), followed by the lack of training on its use and interpretation of the readings (18.7%) [34]. In this context, our findings that being under treatment for hypertension tends to be associated with MNH and that this entity is present in about 10% of subjects who perform an ABPM for different reasons constitute a strong argument for using this tool from time to time in patients already treated under follow-up since otherwise, MNH may not be detected. Moreover, the finding is consistent with the fact that masked hypertension in general is more prevalent in medicated than in untreated subjects [35,36]. Education of primary care physicians on the advantages and disadvantages of both ABPM and HBPM is crucial. The detection of INH and MNH may lead to modifications in the therapeutic strategy since these entities are subtypes of masked hypertension, and it has been established that masked hypertension carries a similar or even higher risk than sustained hypertension [35,36], and many guidelines urge to treat it [37,38].

When analyzing untreated and treated participants separately, the prevalences of INH and MNH were slightly higher in the latter, whereas the associated conditions were similar than when analyzing the whole population. Of course, given the reduced sample size in each subgroup, some of the variables lost statistical significance due to a loss of statistical power.

Finally, our findings must be interpreted in the context of the study limitations: first, the cross-sectional nature of the study precludes cause–effect relationships. Second, patients affiliated with our health plan are mainly urban Argentinean middle-class individuals of European descent who may not represent other ethnicities living in South America. Third, the presence of obstructive sleep apnea, which is a possible cause of nocturnal hypertension, was not assessed. This is an important issue since this condition has been associated with nocturnal hypertension. Fourth, although we have data on sleep duration, we did not evaluate sleep quality, which may have influenced nocturnal BP level; and fifth, the time at which medicated patients took their antihypertensive drugs was not controlled. This may have also influenced nocturnal BP level.

As in any other study, our findings raise questions that must be elucidated in future research: for instance, we only performed one baseline ABPM, and as a consequence, we do not know how reproducible INH and MNH are. Previous studies have suggested that although the reproducibility of nocturnal hypertension is higher than of nocturnal dipping [39], such reproducibility is not optimal, around 33% [4]. The reproducibility of INH and MNH is a topic that warrants future research. Other questions that remain unanswered are the prevalence and clinical impact of these entities in certain subpopulations such as pregnant women and transplant recipients as well as the possible benefit of treating INH and MNH in terms of reducing cardiovascular events. It would also be interesting to collect data on the clinical course of the patients included in this study, and we are currently designing such a protocol.

In conclusion, INH and MNH are frequent entities that are usually overlooked unless properly screened. Since access to ABPM is expensive and limited, determination of clinical associated characteristics such as those detected in this study is critical and might contribute to a more appropriate use of resources.

## Figures and Tables

**Figure 1 diagnostics-13-01419-f001:**
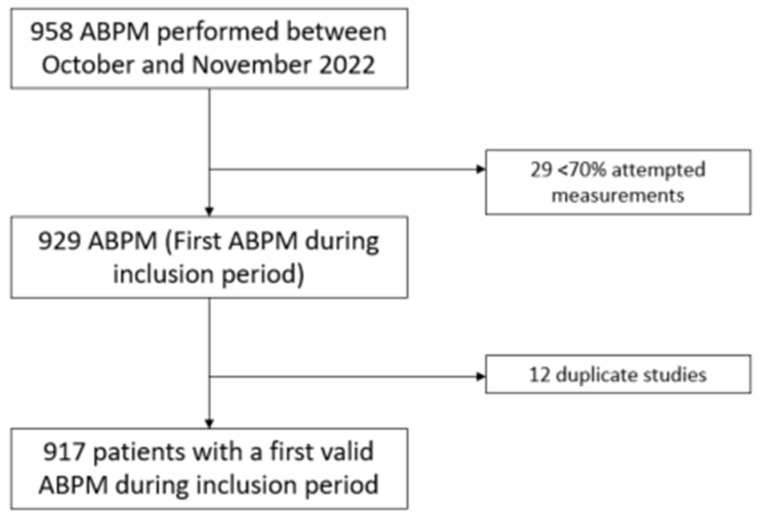
Study population flowchart. ABPM, ambulatory blood pressure monitoring.

**Figure 2 diagnostics-13-01419-f002:**
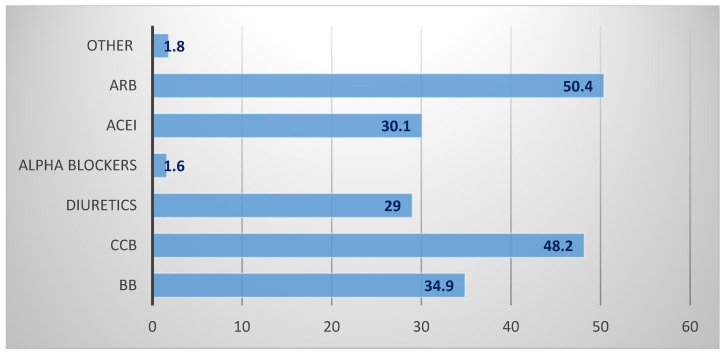
Treatment profile in medicated patients (*n* = 677). ACEI, angiotensin-converting enzyme inhibitors; ARB, angiotensin receptor blockers; BB, beta-blockers; CCB, calcium channel blockers.

**Figure 3 diagnostics-13-01419-f003:**
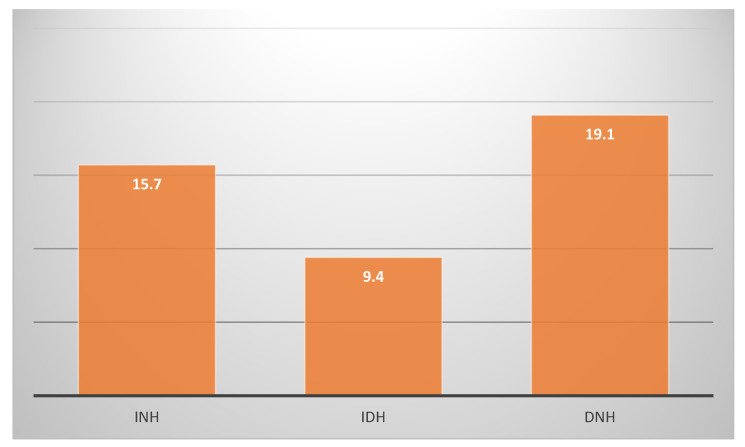
Prevalences of isolated nocturnal hypertension, isolated diurnal hypertension, and day-night hypertension. INH, isolated nocturnal hypertension; IDH, isolated diurnal hypertension; DNH, day-night hypertension.

**Figure 4 diagnostics-13-01419-f004:**
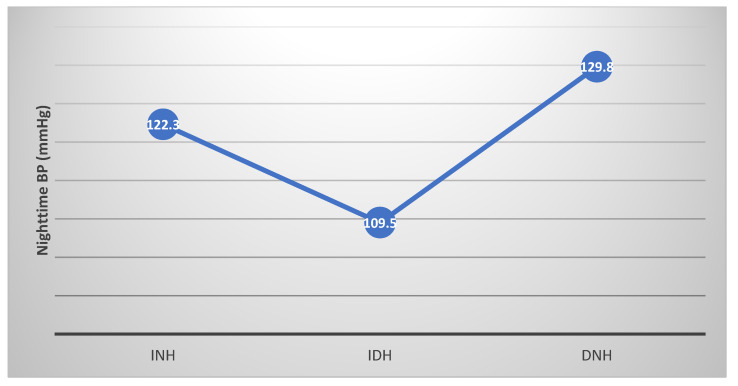
Nighttime systolic blood pressure among the three blood pressure categories: isolated nocturnal hypertension, isolated diurnal hypertension, and day-night hypertension. BP, blood pressure; INH, isolated nocturnal hypertension; IDH, isolated diurnal hypertension; DNH, day-night hypertension. *p* < 0.001 for all three comparisons (one-way ANOVA test with Bonferroni correction).

**Figure 5 diagnostics-13-01419-f005:**
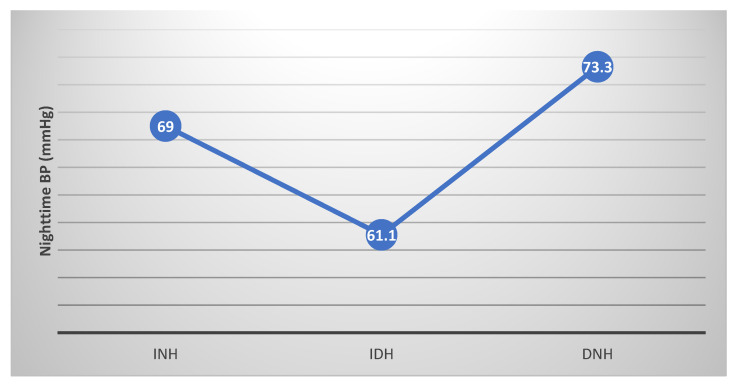
Nighttime diastolic blood pressure among the three blood pressure categories: isolated nocturnal hypertension, isolated diurnal hypertension, and day-night hypertension. BP, blood pressure; INH, isolated nocturnal hypertension; IDH, isolated diurnal hypertension; DNH, day-night hypertension. *p* < 0.001 for all three comparisons (one-way ANOVA test with Bonferroni correction).

**Table 1 diagnostics-13-01419-t001:** Baseline characteristics of study population.

*n*	917
Age, years (SD)	65.1 (14.7)
Male sex, %	43.3
BMI, kg/m^2^ (SD)	28.2 (5.2)
Smoking habits, %	
Current	9.2
Past	16.8
Diabetes, %	11.4
History of ischemic heart disease, %	7.7
History of stroke, %	5.4
Fasting plasma glucose, mg/dL (SD)	100 (18)
Serum creatinine, mg/dL (IQR)	0.87 (0.29)
Total cholesterol, mg/dL (SD)	178 (44.8)
Medicated for hypertension, %	73.9
Mean number of antihypertensive drugs in those medicated (SD) (*n* = 677)	2 (0.9)

BMI, body mass index; IQR, interquartile range; SD, standard deviation.

**Table 2 diagnostics-13-01419-t002:** Blood pressure profile.

*n*	917
Office BP	
Systolic BP, mmHg (SD)	134.6 (17.7)
Diastolic BP, mmHg (SD)	80.4 (12)
Heart rate, bpm (SD)	74.3 (13.3)
Ambulatory BP	
Number of BP measurements (SD)	74.7 (7.4)
Percentage of satisfactory measurements (SD)	94.1 (6.7)
24 h systolic BP, mmHg (SD)	121.9 (12.7)
24 h diastolic BP, mmHg (SD)	71.6 (8.9)
24 h heart rate, bpm (SD)	70.9 (10.3)
Daytime systolic BP, mmHg (SD)	125.9 (12.7)
Daytime diastolic BP, mmHg (SD)	75.1 (9.7)
Daytime heart rate, bpm (SD)	73.5 (11)
Nighttime systolic BP, mmHg (SD)	112.8 (13.8)
Nighttime diastolic BP, mmHg (SD)	63.3 (9.1)
Nighttime heart rate, bpm (SD)	64.5 (9.3)
Duration of sleep, hours (SD)	7.1 (1.6)

BP, blood pressure; bpm, beats per minute; SD, standard deviation.

**Table 3 diagnostics-13-01419-t003:** Comparison between subjects with isolated nocturnal hypertension, isolated diurnal hypertension, day and nighttime hypertension, and day and nighttime normal BP.

Characteristic	INH	IDH	DNH	Normal Day and Nighttime BP	*p*-Value
Age, years (SD)	67.9 (14.7)	62.4 (14.2)	66.1 (14.9)	64.4 (14.6)	0.01
Male sex, %	46.5	51.2	52	38.1	0.003
BMI, kg/m^2^ (SD)	28.5 (4.9)	27.5 (4.5)	28.1 (4.6)	28.3 (5.6)	ns
Smoking habits, %					
Current	4.9	14	12.6	8.4	0.04
Past	13.2	15.1	14.3	18.8	ns
Diabetes, %	16.7	5.9	11.4	10.8	ns
History of ischemic heart disease, %	10.4	7	6.9	7.3	ns
History of stroke, %	8.3	3.5	2.9	5.7	ns
Fasting plasma glucose, mg/dL (SD)	100.8 (15)	97.6 (19.8)	102.7 (22.9)	99.3 (16.6)	ns
Serum creatinine, mg/dL (IQR)	0.99 (0.36)	0.88 (0.26)	1.01 (0.80)	0.98 (0.88)	ns
Total cholesterol, mg/dL (SD)	171.1 (41.5)	184.8 (37.2)	185.7 (45.8)	176.2 (46.2)	0.02
Medicated for hypertension, %	80.6	75.6	77.1	70.7	0.06
Office systolic BP, mmHg (SD)	132.5 (15.3)	144.5 (15.9)	149.2 (18.4)	128.7 (14.7)	<0.001
Office diastolic BP, mmHg (SD)	78.9 (11.1)	84.7 (10.5)	89.1 (12.6)	77.2 (10.6)	<0.001
Office heart rate, bpm (SD)	74 (13.2)	74.2 (12.4)	76.1 (14.4)	73.8 (13)	ns
24 h systolic BP, mmHg (SD)	123.4 (11.2)	128.8 (6.7)	137.7 (9.1)	115.1 (8.7)	<0.001
24 h diastolic BP, mmHg (SD)	72.3 (6.5)	75.3 (7.7)	80.8 (10)	67.7 (6.4)	<0.001
24 h heart rate, bpm (SD)	71.3 (9.8)	70.7 (9.5)	72.9 (12.5)	70.2 (9.7)	0.03
Daytime systolic BP, mmHg (SD)	124.8 (7.2)	136.7 (7.2)	141.3 (9.1)	119.2 (9.5)	<0.001
Daytime diastolic BP, mmHg (SD)	73.8 (7.2)	81.1 (8.5)	84.2 (10.5)	71.3 (7.4)	<0.001
Daytime heart rate, bpm (SD)	73.9 (11.4)	74 (10.6)	75.3 (12.3)	72.6 (10.3)	0.04
Nighttime systolic BP, mmHg (SD)	122.3 (9.2)	109.5 (6.8)	129.8 (12)	104.8 (8.4)	<0.001
Nighttime diastolic BP, mmHg (SD)	68.9 (6.8)	61.1 (5.9)	73.3 (9.6)	58.7 (5.7)	<0.001
Nighttime heart rate, bpm (SD)	65.9 (9.5)	63.3 (8.6)	66 (9.9)	63.8 (9.3)	0.005
Duration of sleep, hours (SD)	7.2 (1.7)	7.1 (1.4)	7.3 (1.6)	7.1 (1.6)	ns

BP, blood pressure; bpm, beats per minute; DNH, day and nighttime hypertension; IDH, isolated daytime hypertension; INH, isolated nocturnal hypertension; ns, non-significant; SD, standard deviation.

**Table 4 diagnostics-13-01419-t004:** Comparison between subjects with and without MNH.

Characteristic	with MNH	without MNH	*p*-Value
Age, years (SD)	66.3 (15.6)	64.9 (14.6)	ns
Male sex, %	49.4	42.7	ns
Smoking habits, %			
Current	5.6	9.6	ns
Past	14.6	17	ns
Diabetes, %	19.1	10.5	0.02
History of ischemic heart disease, %	6.7	7.8	ns
History of stroke, %	7.9	5.1	ns
Medicated for hypertension, %	82	73.1	0.07
Office systolic BP, mmHg (SD)	124.2 (12.1)	135.7 (17.9)	0.01
Office diastolic BP, mmHg (SD)	74.8 (9.7)	81.1 (12.1)	<0.001
Office heart rate, bpm (SD)	74.6 (12.9)	74.3 (13.3)	ns
24 h systolic BP, mmHg (SD)	122 (13.4)	121.9 (12.6)	ns
24 h diastolic BP, mmHg (SD)	71.8 (6.6)	71.6 (9.1)	ns
24 h heart rate, bpm (SD)	72.5 (10.5)	70.7 (10.3)	ns
Daytime systolic BP, mmHg (SD)	123.6 (7.7)	126.2 (13.1)	0.006
Daytime diastolic BP, mmHg (SD)	73.1 (7.5)	75.3 (9.9)	0.01
Daytime heart rate, bpm (SD)	74.9 (11.3)	73.3 (10.9)	ns
Nighttime systolic BP, mmHg (SD)	121.5 (9.1)	111.8 (13.8)	<0.001
Nighttime diastolic BP, mmHg (SD)	68.6 (6.4)	62.7 (9.1)	<0.001
Nighttime heart rate, bpm (SD)	67.1 (10.1)	64.2 (9.2)	0.004

BP, blood pressure; bpm, beats per minute; MNH, masked nocturnal hypertension; ns, non-significant; SD, standard deviation.

**Table 5 diagnostics-13-01419-t005:** Comparison between subjects with isolated nocturnal hypertension, isolated diurnal hypertension, day and nighttime hypertension, and day and nighttime normal BP among untreated participants.

Characteristic	INH	IDH	DNH	Normal Day and Nighttime BP	*p*-Value
Age, years (SD)	58 (15.8)	52.8 (12.2)	58.9 (13.5)	56.1 (15.4)	ns
Male sex, %	53.6	66.7	57.5	32.7	0.001
BMI, kg/m^2^ (SD)	29.2 (5)	26.7 (3.3)	28.4 (4.3)	28.9 (6.9)	ns
Smoking habits, %					
Current	3.6	9.5	15	11.3	ns
Past	10.7	19.1	12.5	14	
Diabetes, %	7.1	4.8	12.5	6.7	ns
History of ischemic heart disease, %	3.6	0	0	1.4	ns
History of stroke, %	0	0	2.5	5.4	ns
Fasting plasma glucose, mg/dL (SD)	100.3 (13.6)	101.5 (29.5)	104.8 (29.3)	95.8 (12.2)	ns
Serum creatinine, mg/dL (IQR)	0.95 (0.27)	0.86 (0.15)	0.92 (0.16)	0.96 (1.24)	ns
Total cholesterol, mg/dL (SD)	201.3 (40.4)	197.8 (38.3)	209.2 (46.2)	196.8 (46.7)	ns
Office systolic BP, mmHg (SD)	131.2 (12.9)	145.2 (14.3)	143.6 (18.4)	128.2 (14.8)	<0.001
Office diastolic BP, mmHg (SD)	80.7 (8.8)	92.1 (8.3)	90.4 (9.8)	79.9 (11)	<0.001
Office heart rate, bpm (SD)	79.3 (13.5)	78 (16)	80.1 (15.4)	77.3 (13.1)	ns
24 h systolic BP, mmHg (SD)	123 (7.3)	126 (5.4)	134.7 (9.9)	114.3 (8.1)	<0.001
24 h diastolic BP, mmHg (SD)	74.6 (4.9)	79.3 (6.4)	82.3 (7.6)	69 (6.1)	<0.001
24 h heart rate, bpm (SD)	74.6 (8.8)	72.3 (9.6)	75.4 (9.8)	72.2 (8.9)	ns
Daytime systolic BP, mmHg (SD)	123.5 (6.8)	133.8 (5.7)	138.7 (9.8)	118.6 (12.4)	<0.001
Daytime diastolic BP, mmHg (SD)	75.6 (5.8)	86 (6.1)	85.9 (7.9)	73.3 (7.1)	<0.001
Daytime heart rate, bpm (SD)	77.7 (10)	75.9 (10.9)	78.7 (11.3)	75 (9.8)	ns
Nighttime systolic BP, mmHg (SD)	121 (12.8)	108 (5.9)	125.9 (11.5)	102.3 (8.6)	<0.001
Nighttime diastolic BP, mmHg (SD)	71.4 (6.1)	63.9 (5.2)	74.2 (7.6)	58.5 (5.3)	<0.001
Nighttime heart rate, bpm (SD)	67.4 (9.5)	64.6 (8.5)	67.6 (8.8)	65.5 (9)	ns
Duration of sleep, hours (SD)	7 (1.8)	7 (1.5)	7.2 (1.7)	7.1 (1.6)	ns

BP, blood pressure; bpm, beats per minute; DNH, day and nighttime hypertension; IDH, isolated daytime hypertension; INH, isolated nocturnal hypertension; IQR, interquartile range; ns, non-significant; SD, standard deviation.

**Table 6 diagnostics-13-01419-t006:** Comparison between subjects with isolated nocturnal hypertension, isolated diurnal hypertension, day and nighttime hypertension, and day and nighttime normal BP among treated participants.

Characteristic	INH	IDH	DNH	Normal Day and Nighttime BP	*p*-Value
Age, years (SD)	70.4 (13.5)	65.5 (13.4)	68.3 (14.8)	67.8 (12.9)	ns
Male sex, %	44.8	46.2	50.4	40.3	ns
BMI, kg/m^2^ (SD)	28.4 (4.9)	27.7 (4.8)	28 (4.7)	28 (4.9)	ns
Smoking habits, %					
Current	5.2	15.4	11.9	7.2	0.04
Past	13.8	13.9	14.8	20.8	ns
Diabetes, %	19	6.3	11.1	12.5	ns
History of ischemic heart disease, %	12.1	9.2	9	9.7	ns
History of stroke, %	10.3	4.6	3	5.8	ns
Fasting plasma glucose, mg/dL (SD)	100.9 (15.4)	96.5 (16.4)	102 (20.6)	100.7 (17.8)	ns
Serum creatinine, mg/dL (IQR)	1 (0.38)	0.89 (0.29)	1.03 (0.91)	0.99 (0.99)	ns
Total cholesterol, mg/dL (SD)	163.4 (38.4)	181.2 (36.3)	178.6 (43.4)	168.3 (43.6)	0.007
Office systolic BP, mmHg (SD)	132.8 (15.8)	144.2 (16.5)	150.8 (18.1)	128.8 (14.7)	<0.001
Office diastolic BP, mmHg (SD)	78.6 (11.5)	82.4 (10.1)	88.8 (13.3)	76 (10.3)	<0.001
Office heart rate, bpm (SD)	72.7 (12.9)	72.9 (10.9)	74.9 (13.9)	72.3 (12.7)	ns
24 h systolic BP, mmHg (SD)	123.5 (12)	129.8 (6.9)	138.6 (8.7)	115.4 (8.9)	<0.001
24 h diastolic BP, mmHg (SD)	71.8 (6.8)	74 (7.7)	80.4 (10.5)	67.2 (6.4)	<0.001
24 h heart rate, bpm (SD)	70.5 (9.9)	70.2 (9.5)	72.1 (13.1)	69.3 (9.9)	ns
Daytime systolic BP, mmHg (SD)	125.1 (7.2)	137.7 (7.4)	142 (8.8)	119.5 (8)	<0.001
Daytime diastolic BP, mmHg (SD)	73.3 (7.5)	79.5 (8.6)	83.7 (11.2)	70.4 (7.3)	<0.001
Daytime heart rate, bpm (SD)	72.9 (11.6)	73.4 (10.5)	74.2 (12.4)	71.7 (10.4)	ns
Nighttime systolic BP, mmHg (SD)	122.6 (8.1)	110 (6.9)	131 (11.9)	105.9 (8)	<0.001
Nighttime diastolic BP, mmHg (SD)	68.4 (6.8)	60.2 (5.8)	73.1 (10.1)	58.8 (5.9)	<0.001
Nighttime heart rate, bpm (SD)	65.5 (9.5)	62.8 (8.6)	65.6 (10.1)	63 (9)	0.008
Duration of sleep, hours (SD)	7.2 (1.6)	7.1 (1.4)	7.4 (1.6)	7 (1.5)	ns
Diuretics, %	32.8	26.2	28.2	28.5	ns
Beta-blockers, %	43.1	26.2	25.9	37.1	0.01
Angiotensin converting enzyme inhibitors, %	31.9	29.2	28.9	30.2	ns
Angiotensin receptor blockers, %	48.3	55.4	55.6	48.2	ns
Calcium channel blockers, %	44.8	49.2	43	51	ns
Alpha blockers, %	3.5	0	1.5	1.4	ns
Other, %	0.9	1.5	2.2	1.9	ns

BP, blood pressure; bpm, beats per minute; DNH, day and nighttime hypertension; IDH, isolated daytime hypertension; INH, isolated nocturnal hypertension; IQR, interquartile range; ns, non-significant; SD, standard deviation.

**Table 7 diagnostics-13-01419-t007:** Comparison between subjects with and without MNH among untreated participants.

Characteristic	with MNH	without MNH	*p*-Value
Age, years (SD)	56 (18)	56.6 (14.7)	ns
Male sex, %	62.5	40.8	ns
Smoking habits, %			
Current	0	11.7	ns
Past	12.5	13.9	ns
Diabetes, %	0	8.1	ns
History of ischemic heart disease, %	0	1.4	ns
History of stroke, %	0	4.1	ns
Office systolic BP, mmHg (SD)	123.8 (9.1)	133.3 (16.8)	0.02
Office diastolic BP, mmHg (SD)	75.4 (6.3)	83.4 (11.4)	0.007
Office heart rate, bpm (SD)	80.9 (12.8)	77.9 (13.9)	ns
24 h systolic BP, mmHg (SD)	122.9 (7.1)	119.6 (11.5)	ns
24 h diastolic BP, mmHg (SD)	73.6 (3.9)	72.7 (8.5)	ns
24 h heart rate, bpm (SD)	75.3 (10.1)	72.9 (9.1)	ns
Daytime systolic BP, mmHg (SD)	122.8 (6.3)	123.9 (13.9)	ns
Daytime diastolic BP, mmHg (SD)	74.7 (4.9)	76.9 (9)	ns
Daytime heart rate, bpm (SD)	79.1 (11.6)	75.8 (10.8)	ns
Nighttime systolic BP, mmHg (SD)	121.9 (14)	108 (13)	<0.001
Nighttime diastolic BP, mmHg (SD)	70.6 (6.6)	62.6 (8.6)	0.003
Nighttime heart rate, bpm (SD)	67.6 (9.6)	65.9 (8.9)	ns

BP, blood pressure; bpm, beats per minute; MNH, masked nocturnal hypertension; ns, non-significant; SD, standard deviation.

**Table 8 diagnostics-13-01419-t008:** Comparison between subjects with and without MNH among treated participants.

Characteristic	with MNH	without MNH	*p*-Value
Age, years (SD)	68.6 (14.2)	68.1 (13.4)	ns
Male sex, %	46.6	43.3	ns
Smoking habits, %			
Current	6.9	8.8	ns
Past	15.1	18.1	ns
Diabetes, %	23.3	11.4	0.004
History of ischemic heart disease, %	8.2	10.1	ns
History of stroke, %	9.6	5.5	ns
Office systolic BP, mmHg (SD)	124.3 (12.7)	136.7 (18.2)	<0.001
Office diastolic BP, mmHg (SD)	74.6 (10.4)	80.2 (12.2)	<0.001
Office heart rate, bpm (SD)	73.2 (12.6)	72.9 (12.9)	ns
24 h systolic BP, mmHg (SD)	121.8 (14.4)	122.9 (12.9)	ns
24 h diastolic BP, mmHg (SD)	71.3 (7)	71.2 (9.4)	ns
24 h heart rate, bpm (SD)	71.9 (10.5)	69.9 (10.6)	ns
Daytime systolic BP, mmHg (SD)	123.8 (8)	127 (12.7)	0.03
Daytime diastolic BP, mmHg (SD)	72.7 (7.9)	74.6 (10.1)	ns
Daytime heart rate, bpm (SD)	74.1 (11.1)	72.4 (11.1)	ns
Nighttime systolic BP, mmHg (SD)	121.4 (7.8)	113.3 (13.9)	<0.001
Nighttime diastolic BP, mmHg (SD)	68.1 (6.4)	62.8 (9.4)	<0.001
Nighttime heart rate, bpm (SD)	67 (10.3)	63.6 (9.2)	0.003

BP, blood pressure; bpm, beats per minute; MNH, masked nocturnal hypertension; ns, non-significant; SD, standard deviation.

## Data Availability

Additional data are available from the corresponding author on reasonable request.

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
