# Peer review of "Prevalence and Characteristics of Isolated Nocturnal Hypertension and Masked Nocturnal Hypertension in a Tertiary Hospital in the City of Buenos Aires"

_diagnostics, 2023, doi:10.3390/diagnostics13081419_

Round 1
Reviewer 1 Report
In the present study, the authors report on the prevalence of INH and MNH in a hypertensive population from Buenos Aires trying to identify the predictors of these conditions which actually were a few clinical variables of little practical impact.
This reviewer has the following comments and suggestions.
The introduction is too long. The authors should come to the point without beating around the bush. The issue of low-income countries for instance is irrelevant to the present data. The introduction should be shortened by at least one third. References 8-10 should be updated.
The selection criteria should be better described. Did all subjects included have a BP>= 140/90 mmHg at enrolment and/or were they on antihypertensive treatment? The present results pertain only to a hypertensive sample.
The authors say that the sample size was estimated assuming a prevalence of INH of 12.9% and that the minimum number of patients to be recruited was calculated in 480. To prove what? The aim of the study was only to investigate the prevalence of the conditions.
¾ of participants were on treatment. The prevalence of the conditions should be given for true MH and for MUCH, separately.
Patients with stroke or AMI should be excluded because these conditions can affect the 24h BP rhythms.
It is unclear with which subjects participants with INH or MNH were compared (numbers are not reported in tables and figures). INH subjects should be compared with people with both normal day and night-time BP and with subjects with IDH (separately). People with DNH also have high night-time BP and thus should be excluded from the comparisons or treated as a distinct group.
For the predictor issue, I am afraid that the authors have too few variables to include in the regressions and that they should limit their investigation to the prevalence of the conditions and to between-group differences.
Author Response
Reviewer #1
In the present study, the authors report on the prevalence of INH and MNH in a hypertensive population from Buenos Aires trying to identify the predictors of these conditions which actually were a few clinical variables of little practical impact.
This reviewer has the following comments and suggestions.
The introduction is too long. The authors should come to the point without beating around the bush. The issue of low-income countries for instance is irrelevant to the present data. The introduction should be shortened by at least one third. References 8-10 should be updated.
Answer: we have shortened the Introduction section, according to the reviewer´s suggestion. As a consequence, references 8-10 have been removed (pages 1 and 2, highlighted in yellow).
The selection criteria should be better described. Did all subjects included have a BP>= 140/90 mmHg at enrolment and/or were they on antihypertensive treatment? The present results pertain only to a hypertensive sample.
Answer: we consecutively included patients that performed an ABPM as prescribed by their treating physician to diagnose hypertension or to assess its control. Therefore, subjects could have any BP level and be or not under antihypertensive treatment. This piece of information has now been added to the Methods section (page 2, highlighted in yellow).
The authors say that the sample size was estimated assuming a prevalence of INH of 12.9% and that the minimum number of patients to be recruited was calculated in 480. To prove what? The aim of the study was only to investigate the prevalence of the conditions.
Answer: sample size estimation is not only recommended when testing a hypothesis, but also to estimate proportions with a specified level of confidence and precision.1
- https://epitools.ausvet.com.au/oneproportion.
¾ of participants were on treatment. The prevalence of the conditions should be given for true MH and for MUCH, separately.
Answer: the prevalence of INH was 17.1% (95%CI 14.5-20.1%) in treated and 11.7% (95%CI 8.2-16.5%) in untreated subjects, whereas the prevalence of MNH was 10.8% (95%CI 8.6-13.3%) in treated and 6.7% (95%CI 4.1-10.7%) in untreated subjects. This piece of information has now been added to the Results section (page 9, highlighted in yellow).
Patients with stroke or AMI should be excluded because these conditions can affect the 24h BP rhythms.
Answer: when performing a sensitivity analysis, excluding subjects with a history of cardio or cerebrovascular disease, the results did not significantly change: the prevalence of INH was 15% (95%CI 12.7-17.6%) and the prevalence of MNH was 9.6% (95%CI 7.7-11.8%). This piece of information has now been added to the Results section (page 5, highlighted in yellow).
It is unclear with which subjects participants with INH or MNH were compared (numbers are not reported in tables and figures). INH subjects should be compared with people with both normal day and night-time BP and with subjects with IDH (separately). People with DNH also have high night-time BP and thus should be excluded from the comparisons or treated as a distinct group.
Answer: we have changed our analysis and now present the comparison among the four groups: INH, IDH, DNH and normal day/nighttime BP (page 6, highlighted in yellow and tables 3, 5 and 6).
For the predictor issue, I am afraid that the authors have too few variables to include in the regressions and that they should limit their investigation to the prevalence of the conditions and to between-group differences.
Answer: according to the reviewer´s suggestion we have now removed the regression analyses and present only the prevalence sans the comparison between groups.

Reviewer 2 Report
The authors performed retrospective analysis of almost 1,000 hypertensive patients undergoing ABPM to detect nocturnal hypertension (both isolated and masked nocturnal hypertension). They found a high prevalence. Unfortunately, no specific predictor for nocturnal hypertension was found that can be depicted without performing ABPM.
The paper is well written with adequate statistics and references. However, there are two major limitations of the study: First, there is no information about obstructive sleep apnoea syndrome (OSAS). This disease is probably the main cause of most patients with nocturnal hypertension. I recommend that authors collect data about the presence of OSAS in the patient cohort. Otherwise, paper would be very difficult to interpret. Secondly, there is no information about the clinical course of the patients. Follow up would dramatically increase the importance of the study.
Minor comments:
- How were the statistical analyses performed?
- Figures are of low quality. Figure 2 may be sorted.
- Was there an association of specific antihypertensive drug in take with nocturnal hypertension?
Author Response
Reviewer #2:
The authors performed retrospective analysis of almost 1,000 hypertensive patients undergoing ABPM to detect nocturnal hypertension (both isolated and masked nocturnal hypertension). They found a high prevalence. Unfortunately, no specific predictor for nocturnal hypertension was found that can be depicted without performing ABPM.
The paper is well written with adequate statistics and references. However, there are two major limitations of the study: First, there is no information about obstructive sleep apnoea syndrome (OSAS). This disease is probably the main cause of most patients with nocturnal hypertension. I recommend that authors collect data about the presence of OSAS in the patient cohort. Otherwise, paper would be very difficult to interpret. Secondly, there is no information about the clinical course of the patients. Follow up would dramatically increase the importance of the study.
Answer: we agree with the reviewer on that information on OSAS would improve and enrich the manuscript. Unfortunately, we do not have information on that issue. We have now acknowledged this in the limitations paragraph in the Discusion section (pages 15 and 16 , highlighted in yellow).
Regarding follow up, the present study was conceived as a cross-sectional one, aiming at detecting prevalences. Cross sectional studies, by definition, do not include follow up. We agree with the reviewer that information on the clinical course of patients is important, and we have now stated this fact in the perspectives paragraph in the Discusion section (page 16, highlighted in yellow).
Minor comments:
- How were the statistical analyses performed?
Answer: statistical analyses were performed using STATA 14 (StataCorp, LLC). This piece of information has now been added to the Methods section (page 3, highlighted in yellow).
- Figures are of low quality. Figure 2 may be sorted.
Answer: the quality of Figure 2 has been improved.
- Was there an association of specific antihypertensive drug in take with nocturnal hypertension?
Answer: we have now provided a separate analysis for treated and untreated subjects, analyzing associations between antihypertensive drug treatment and INH in the former (pages 9-14, highlighted in yellow).

Round 2
Reviewer 1 Report
No more comments
Author Response
No more comments.
Reviewer 2 Report
All comments have been adequately addressed.
Author Response
No more comments.